# Radical asymmetric intramolecular α-cyclopropanation of aldehydes towards bicyclo [3.1.0]hexanes containing vicinal all-carbon quaternary stereocenters

Liu Ye[1], Qiang-Shuai Gu[1], Yu Tian[1], Xiang Meng[1], Guo-Cong Chen[1] & Xin-Yuan Liu [1]

The development of a general catalytic method for the direct and stereoselective construction of cyclopropanes bearing highly congested vicinal all-carbon quaternary stereocenters remains a formidable challenge in chemical synthesis. Here, we report an intramolecular radical cyclopropanation of unactivated alkenes with simple α-methylene group of aldehydes as C1 source via a Cu(I)/secondary amine cooperative catalyst, which enables the single-step construction of bicyclo[3.1.0]hexane skeletons with excellent efficiency, broad substrate scope covering various terminal, internal alkenes as well as diverse (hetero)aromatic, alkenyl, alkyl-substituted geminal alkenes. Moreover, this reaction has been successfully realized to an asymmetric transformation, providing an attractive approach for the construction of enantioenriched bicyclo[3.1.0]hexanes bearing two crucial vicinal all-carbon quaternary stereocenters with good to excellent enantioselectivity. The utility of this method is illustrated by facile transformations of the products into various useful chiral synthetic intermediates. Preliminary mechanistic studies support a stepwise radical process for this formal [2 + 1] cycloaddition.

[1] Department of Chemistry, South University of Science and Technology of China 518055 Shenzhen, China. Liu Ye and Qiang-Shuai Gu contributed equally to this work. Correspondence and requests for materials should be addressed to X.-Y.L. (email: liuxy3@sustc.edu.cn)

Chiral bicyclo[3.1.0]hexanes bearing one or more all-carbon quaternary stereocenters are significant structural motifs occurring in a large number of natural and unnatural compounds with important biological activities (Fig. 1a)[1–6]. In particular, such skeletons have also been widely applied as highly useful chiral building blocks in organic synthesis because of unique chemical reactivity for fragmentation and rearrangement[7–10]. Various approaches to access these structurally unique scaffolds have been developed[11–18], and most of them are based on the asymmetric intramolecular cyclopropanation of olefins with metallocarbenes as the C1 component[19–27]. Despite these significant achievements in the field of metallocarbene chemistry, reactive prefunctionalized reagents, such as diazos, sulfonyl hydrazones, and ylides, have been mostly used as the metallo-carbene precursors as the C1 component in this system[11–27]. On the other hand, it is well-known that the efficient construction of chiral all-carbon quaternary stereocenter generally represents a significant and highly important task, but is among the most challenging objectives in organic synthesis due to the inherently unfavorable steric hindrance and relatively small steric differences for efficient enantiocontrol[28–31]. Noteworthy is that the efficient formation of bicyclo[3.1.0]hexane scaffolds containing two sterically congested vicinal all-carbon quaternary stereocenters with conventional metallocarbene strategies remains a formidable challenge[19–27]. To circumvent the aforementioned challenges, the invention of a catalytic enantioselective intramolecular cyclopropanation method capable of constructing structurally diverse bicyclo[3.1.0]hexane skeletons containing vicinal all-carbon quaternary stereocenters, preferably by using readily available and simple methylene group as C1 source, is highly desirable and will be of great synthetic importance.

Recently, impressive progress has been achieved in the development of intermolecular cyclopropanation of olefins using simple methyl group as C1 source[32–35], which has obvious advantages over reactive prefunctionalized precursors with respect to availability of the starting materials, operation safety, environmental benignity, and atom economy. In particular, Antonchick and co-workers reported a seminal work on copper (I)-catalyzed intermolecular [2 + 1] cycloaddition of electron-deficient alkenes with the methyl group in aryl methyl ketones as C1 source for the construction of cyclopropanes with good efficiency through a radical process[33]. Compared with these attractive racemic attributes, the development of catalytic asymmetric variant of this type of reaction remains a formidable unexplored challenge, which might be attributed to the relatively harsh reaction conditions (at 110 °C) and the highly reactive nature of the involved radical species[36–43]. With our continuing interest in developing the challenging asymmetric radical reactions with the dual-catalytic system through the combination of transition metal catalysis and organocatalysis[44–48], we became interested in employing Cu(I)/chiral amine cooperative catalysis[49–54] for realizing the asymmetric radical intramolecular cyclopropanation of alkenes with a simple α-methylene of aldehydes for the efficient construction of structurally diverse bicyclo[3.1.0]hexane skeletons containing two crucial vicinal all-carbon quaternary stereocenters.

In this scenario, we envisaged that the enamine intermediate, in situ generated from a chiral secondary amine with an aldehyde of the rationally designed alkenyl aldehyde substrate 1, could undergo a selective single electron transfer (SET)[55–59], followed by 6-endo-trig cyclization and cyclopropanation to afford the optically enriched bicyclo[3.1.0]hexane motif with

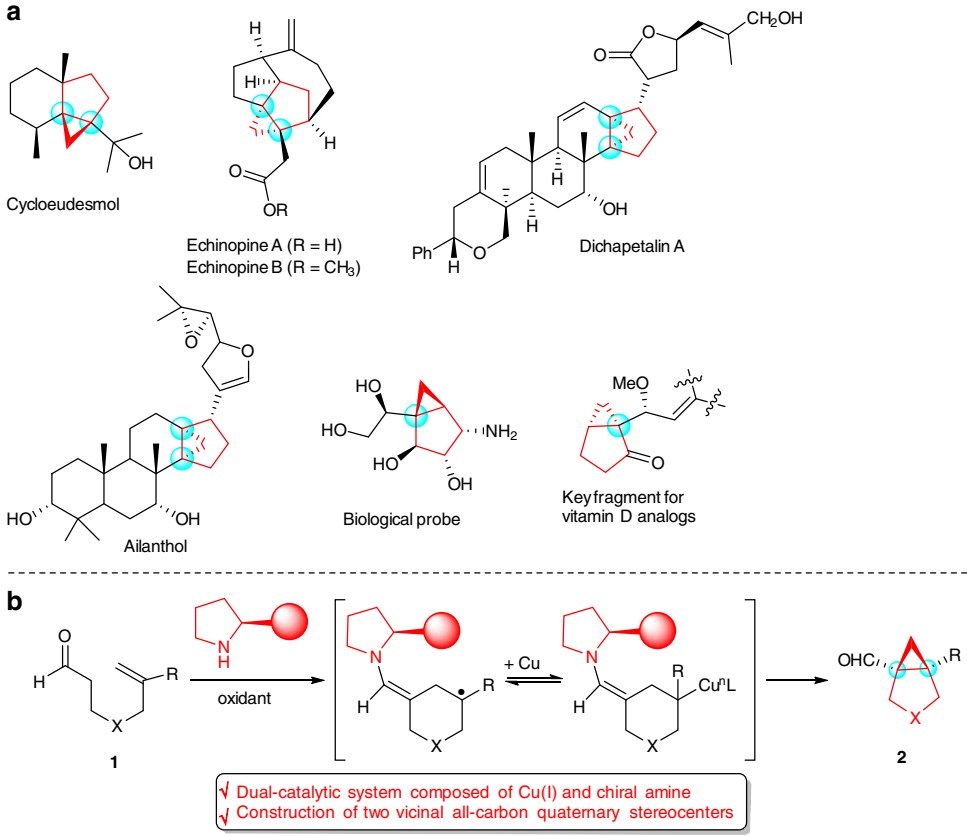

**Fig. 1** Bicyclo[3.1.0]hexane skeletons-containing compounds and our synthetic proposal. **a** Representative natural and unnatural products containing bicyclo[3.1.0]hexanes bearing quaternary stereocenters. **b** Our envisioned catalytic asymmetric radical cyclopropanation of alkenyl aldehyde

**Table 1 Optimization of reaction conditions**

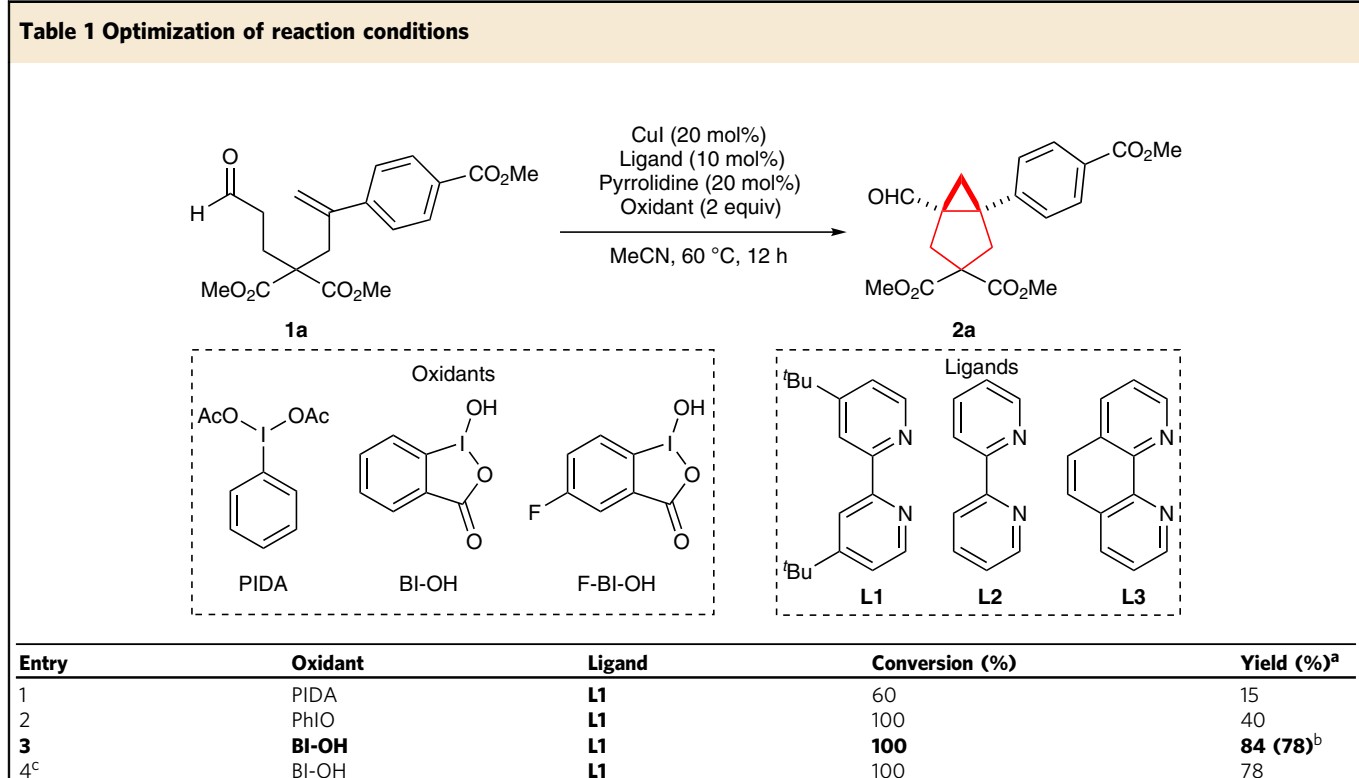

| Entry | Oxidant | Ligand | Conversion (%) | Yield (%)[a] |
|---|---|---|---|---|
| 1 | PIDA | L1 | 60 | 15 |
| 2 | PhIO | L1 | 100 | 40 |
| **3** | **BI-OH** | L1 | **100** | **84 (78)**[b] |
| 4[c] | BI-OH | L1 | 100 | 78 |
| 5 | F-BI-OH | L1 | 100 | 75 |
| 6 | BI-OH | — | 85 | 40 |
| 7[d] | BI-OH | L1 | 100 | 68 |
| 8 | BI-OH | L2 | 100 | 83 |
| 9 | BI-OH | L3 | 100 | 80 |
| 10[e] | BI-OH | L1 | 80 | 30 |
| 11[f] | BI-OH | L1 | 90 | Trace |
| 12[g] | BI-OH | L1 | 100 | 74 |
| 13[h] | BI-OH | L1 | 100 | 44 |

Reactions were performed on 0.1 mmol scale
[a] Yield was determined by crude $^1$H NMR using $CH_2Br_2$ as internal standard
[b] Isolated yield
[c] $T$ 80 °C, 8 h
[d] **L1** (20 mol%)
[e] Without pyrrolidine
[f] Without CuI
[g] CuBr (20 mol%) was used
[h] Cu(OAc)$_2$ (20 mol%) was used

Cu(I)/chiral amine cooperative catalyst. Noteworthy is that Huang and co-workers have recently reported an asymmetric intramolecular α-cyclopropanation of alkenyl aldehydes with the in situ-generated α-iodoaldehyde as a donor/acceptor carbene mimetic, invoking a stepwise double electrophilic alkylation cascade through *5-exo-trig* cyclization[27]. In this reaction, the bis-alkyl substituents at the double bond were essential for implementing the enantioselective reaction, possibly due to indispensable formation of carbocation intermediates, with an exceptionally stoichiometric amount of chiral secondary amine as the promoter. We report herein the successful development of a cooperative catalytic system composed of Cu(I) and chiral secondary amine, which enables the asymmetric radical cyclopropanation of unactivated alkenes using simple α-methylene group of aldehydes as a C1 source, providing the single-step construction of fundamental yet synthetically formidable enantioenriched bicyclo[3.1.0]hexane skeletons bearing two crucial vicinal all-carbon quaternary stereocenters with good to excellent enantioselectivity.

## Results

**Racemic radical intramolecular cyclopropanation**. To probe the feasibility of our proposed assumption (Fig. 1b), we initiated our investigation of searching for a suitable cooperative catalytic system for the development of a non-stereoselective cyclopropanation of alkenyl aldehyde **1** using simple α-methylene group of aldehyde as a C1 source (Table 1). As such, we examined the reaction of **1a** with the combination of pyrrolidine and copper iodide in the presence of PhI(OAc)$_2$ as the oxidant and the desired product **2a** was observed, albeit only in 15% yield (entry 1, Table 1). Encouraged by this result, we then systematically optimized the reaction parameters, and found that the oxidants (entries 2–5, Table 1) as well as metal ligands (entries 6–9, Table 1) have significant impact on the reaction efficiency and selectivity. In particular, among the oxidants screened, cyclic hypervalent iodine(III) reagents, such as BI-OH worked well for this transformation[60]. Notably, the reaction yield was dramatically reduced in the absence of pyrrolidine (entry 10, Table 1) and only trace amount of product was detected without copper salt

**Table 2 Scope for substrates bearing aromatic and heterocyclic rings of non-stereoselective reaction**

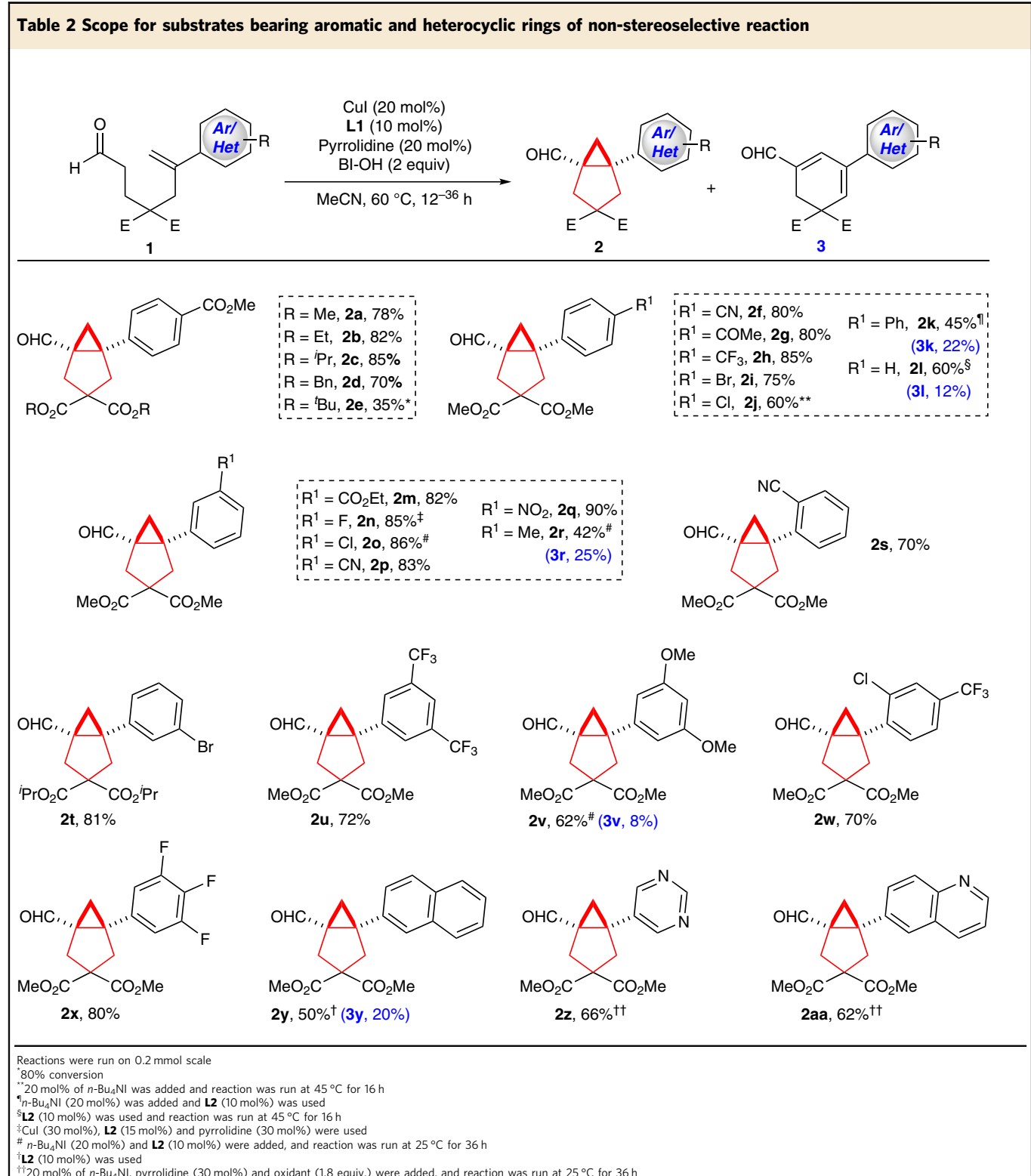

Reactions were run on 0.2 mmol scale
*80% conversion
**20 mol% of n-Bu₄NI was added and reaction was run at 45 °C for 16 h
¶n-Bu₄NI (20 mol%) was added and L2 (10 mol%) was used
§L2 (10 mol%) was used and reaction was run at 45 °C for 16 h
‡CuI (30 mol%), L2 (15 mol%) and pyrrolidine (30 mol%) were used
# n-Bu₄NI (20 mol%) and L2 (10 mol%) were added, and reaction was run at 25 °C for 36 h
†L2 (10 mol%) was used
††20 mol% of n-Bu₄NI, pyrrolidine (30 mol%) and oxidant (1.8 equiv.) were added, and reaction was run at 25 °C for 36 h

(entry 11, Table 1), revealing that synergistic combination of copper and aminocatalyst is indispensable for the cyclopropanation reaction. Other copper salts were also screened, while gave inferior results (entries 12–13, Table 1). Finally, the optimal conditions were identified as 20 mol% of CuI and pyrrolidine in the presence of 2 equiv. of BI-OH and 10 mol% of **L1** in MeCN at 60 °C for 12 h, providing the desired product **2a** in 78% isolated yield.

With the optimal conditions being established, we next investigated the scope of this intramolecular cyclopropanation in racemic form and the results are summarized in Table 2 and Table 3. First, a series of substrates bearing different malonate-tethered groups were investigated. The results revealed that different substituents have a negligible effect on the reaction efficiency to afford the corresponding products **2a–2d** in 70–85% yields except for the bulky di-*tert*-butyl ester-tethered product **2e**

**Table 3 Other types of substrates of non-stereoselective reaction**

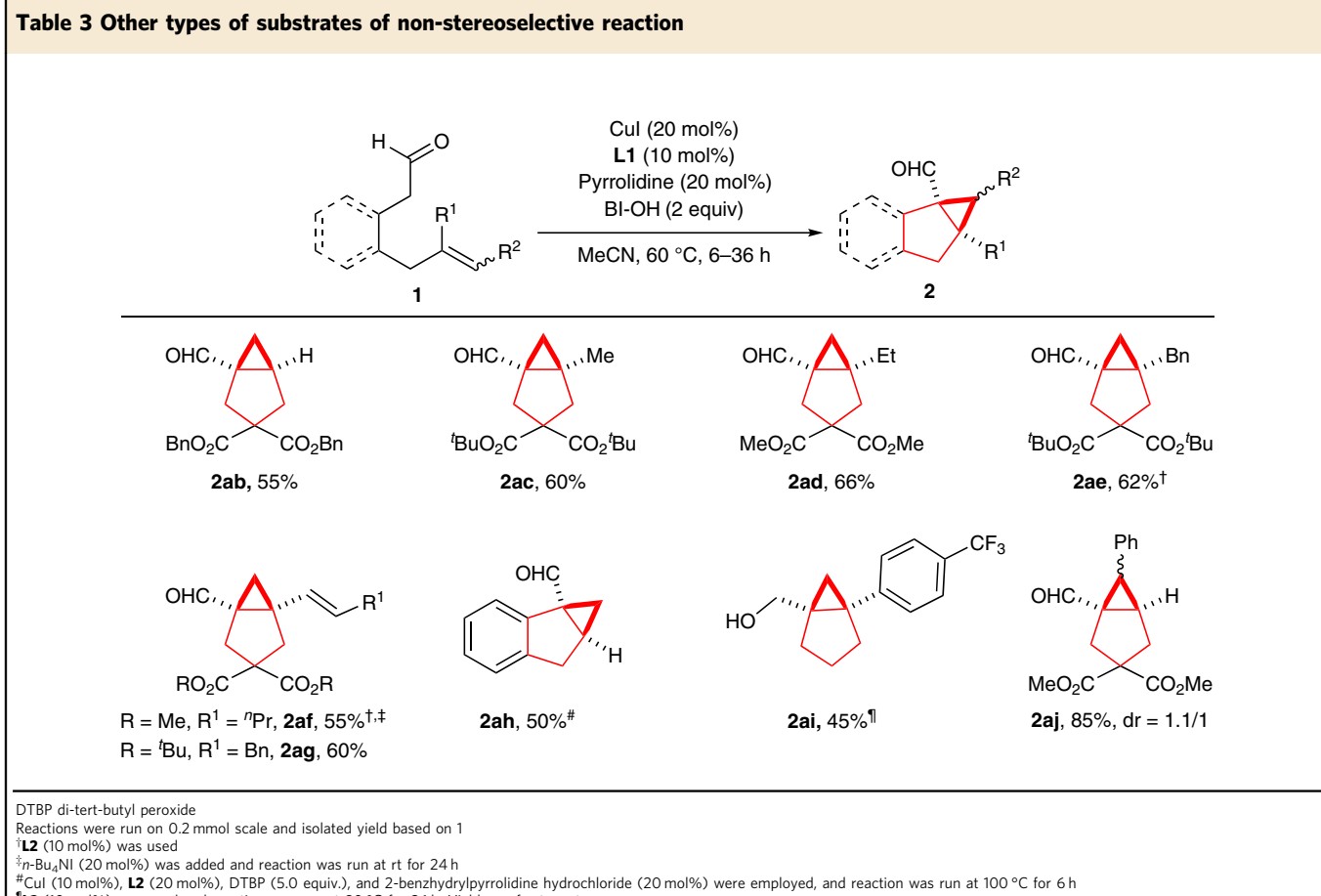

DTBP di-tert-butyl peroxide
Reactions were run on 0.2 mmol scale and isolated yield based on 1
[†] L2 (10 mol%) was used
[‡] n-Bu$_4$NI (20 mol%) was added and reaction was run at rt for 24 h
[#] CuI (10 mol%), L2 (20 mol%), DTBP (5.0 equiv.), and 2-benzhydrylpyrrolidine hydrochloride (20 mol%) were employed, and reaction was run at 100 °C for 6 h
[¶] L2 (10 mol%) was used and reaction was run at 80 °C for 36 h. Yield was for two steps

(35%). Furthermore, a range of diversely functionalized alkenyl aldehydes **1**, including those having aryl groups with electron-withdrawing (CN, COMe, halides, CO$_2$Et, CF$_3$, or NO$_2$) were found to be suitable substrates to effectively convert into the bicyclic products in 42–90% yields, irrespective of the position of these substituents on the aromatic ring and substitution pattern (**2f–2x**). It was interesting to find that substrates bearing electron-donating substituents (*meta*-Me, or *meta*-OMe) or electron-neutral (H) arene rings were all suitable for the reaction to produce the desired products **2l**, **2r**, and **2v** in moderate to good yields, while with a small amount of unexpected 1,3-cyclohex-adiene derivatives **3**. It should be noted that the addition of a catalytic amount of *tetra*-(n-butyl)ammonium iodide (n-Bu$_4$NI) could improve the reaction efficiency of **1j**, **1k**, **1o**, **1r**, and **1v** significantly, considering the observation that trace amount or low yields of products were obtained for these substrates under the standard conditions. In addition, this reaction shows good compatibility with fused aromatic 2-naphthyl-substituted alkene, giving the corresponding product **2y** in 50% yield, along with **3y** in 20% yield. Given the importance of heterocyclic structures in the synthesis of biologically important molecules, we were pleased to find that substrates containing heterocyclic structures such as pyrimidine or quinolone, readily participated in the reaction to give products **2z** and **2aa** in 66 and 62% yields, respectively. It should be noted that many functional groups, such as halides (**2i–j**, **2n**), ester (**2m**), ketone (**2g**), nitrile (**2f**, **2p**), and even nitro (**2q**) as well as heterocycles (**2z** and **2aa**) were all compatible under these conditions. These features indicate that this general cyclopropanation reaction exhibits great functional group tolerance to offer versatile opportunities for further useful modifications, highlighting the generality of this transformation.

As an extension of the above cyclopropanation reaction, more challenging substrates with mono-substituted alkene (**1ab**) and alkyl-substituted alkenes (**1ac–1ae**) could also be employed in the reaction (Table 3). Under the conditions identical to those of cyclopropanation detailed above, all of them exhibited good reactivity. To further demonstrate the synthetic utility of the reaction, we tested more complex diene substrates to afford the desired products **2af** and **2ag** containing alkenyl group in synthetically useful yields, respectively. These results clearly demonstrated that this reaction would broadly expand the application scope of this radical cyclopropanation strategy, in that previous reports have been limited to only activated alkenes and styrene-type alkenes[32,33]. Moreover, the phenyl-tethered substrate **1ah** also proved to be a suitable substrate, providing tricyclic product **2ah** in 50% yield at a higher temperature. Unlike tethered substrates, the use of linear substrates without the Thorpe-Ingold effects is generally far less studied, probably due to the unfavorable entropy factor and proximity effects in the cyclic transition state of such processes[61]. It is more encouraging to note that the linear substrate **1ai** was also applicable to this process, affording **2ai** in 45% yield under the similar conditions after the reduction with NaBH$_4$. It is noteworthy that internal alkene substrate **1aj** was also compatible to give **2aj** as a 1.1:1 mixture of diastereomers in 85% yield.

**Asymmetric radical intramolecular cyclopropanation**. Having established the proof-of-principle for the intramolecular α-cyclopropanation of aldehydes, we thus switched our attention on the challenging asymmetric α-cyclopropanation of aldehydes. Our investigation began with the evaluation of a series of chiral

**Table 4 Optimization of asymmetric reaction**

| Entry | Amine | Oxidant | T (°C) | Conversion (%) | er[a] |
|-------|-------|---------|--------|----------------|------|
| 1 | A12 | BI-OH | 60 | 100 | 82.5:17.5 |
| 2 | A12 | BI-OH | 30 | 80 | 85.5:14.5 |
| 3 | A12 | F-BI-OH | 30 | 100 | 88:12 |
| 4 | A12 | F-BI-OH | 20 | 85 | 91.5:8.5 |
| 5 | A12 | DF-BI-OH | 20 | 100 | 90.5:9.5 |
| 6 | A13 | F-BI-OH | 20 | 90 | 91:9 |
| 7 | A14 | F-BI-OH | 20 | 90 | 90:10 |
| 8 | A15 | F-BI-OH | 20 | 100 | 70:30 |
| 9[b] | A12 | DF-BI-OH | 10 | 50 | 92.5:7.5 |
| 10[c] | A12 | DF-BI-OH | 10 | 100 | 93:7 |
| **11[d]** | **A14** | **DF-BI-OH** | **10** | **100** | **95:5** |

Reactions were run on 0.05 mmol scale
[a]Determined by chiral stationary HPLC
[b]96 h
[c]20 mol% of $n$-Bu$_4$NI was added and reaction time was 72 h
[d]20 mol% of $n$-Bu$_4$NI was added and 60% isolated yield of **2a** after 72 h

secondary amine catalysts (for details, see Supplementary Tables 1–6). To our disappointment, the imidazolidinone catalyst and most commercially available chiral secondary amines were all ineffective. After a thorough evaluation of different Hayashi-Jørgensen's organocatalysts[62], we were grateful to find that **A12** with two bulky *tert*-butyl substituents at the *meta* positions was effective, affording good enantioselectivity (82.5:17.5 er, entry 1, Table 4 and Supplementary Table 1). After systematic optimization efforts, we found that various reaction parameters were crucial for obtaining the good result. Remarkable solvent and ligand effects were observed in this transformation and ligand **L1** with CH$_3$CN as the solvent was the best in terms of enantioselectivity (Supplementary Tables 2 and 3). The enantioselectivity was greatly affected by the reaction temperature and a significantly increased enantioselectivity (91.5:8.5 er) was obtained by lowering the reaction temperature to 20 °C (entry 4, Table 4 and Supplementary Table 4). While the reaction rate was very slow at 20 °C with BI-OH as the oxidant, the choice of a stronger oxidant F-BI-OH to accelerate the reaction rate at low temperature was significant for the full conversion of **1a** (entries 2–5, Table 4 and Supplementary Table 5). Varying the size of the silicon group of diarylprolinol silyl ethers had also a profound influence on the stereoselectivity and the bulky silyl ethers (**A12–A14**) all resulted in good enantiomeric excess (90:10 to 91.5:8.5 er) (entries 4–8). Further lowering the reaction temperature to 10 °C could gave a slightly better er, but only 50%

of conversion was observed after 96 h even in the presence of a much stronger oxidant DF-BI-OH (entry 9). Further investigation revealed that the addition of a catalytic amount of ammonium salt could improve reaction efficiency remarkably and $n$-Bu$_4$NI gave the best results (up to 60% yield and 95:5 er, entries 10 and 11, and Supplementary Table 6) after screening a variety of additives, which is largely due to its capacity of improving the solubility of insoluble oxidant in the organic solvent.

Under the optimized conditions in hand, the generality of the current enantioselective intramolecular radical cyclopropanation reaction was next investigated (Table 5). First, a wide range of substrates bearing differently malonate-tethered groups were surveyed to smoothly deliver the desired products **2a–2d** bearing two contiguous all-carbon quaternary stereocenters in 43–60% yields with 92.5:7.5 to 95:5 er. It was found that both the position and electronic nature of the substituents on the aromatic ring have a negligible influence on the reaction efficiency and stereoselectivity of the process. For example, substrates bearing a series of functional groups (CN, COMe, halides, CO$_2$Et, CF$_3$, or NO$_2$) at different positions (*para* or *meta*) of the aryl ring reacted smoothly to afford the corresponding products **2f–2q** with moderate to good yields and good to excellent levels of enantioselectivity. Moreover, the sterically hindered *ortho*-substituted substrate **1s** also provided the desired product **2s** in 62% yield and 93:7 er. In sharp contrast to the previous works[32–35], the mild reaction conditions make this asymmetric

**Table 5 Substrate scope for asymmetric reaction**

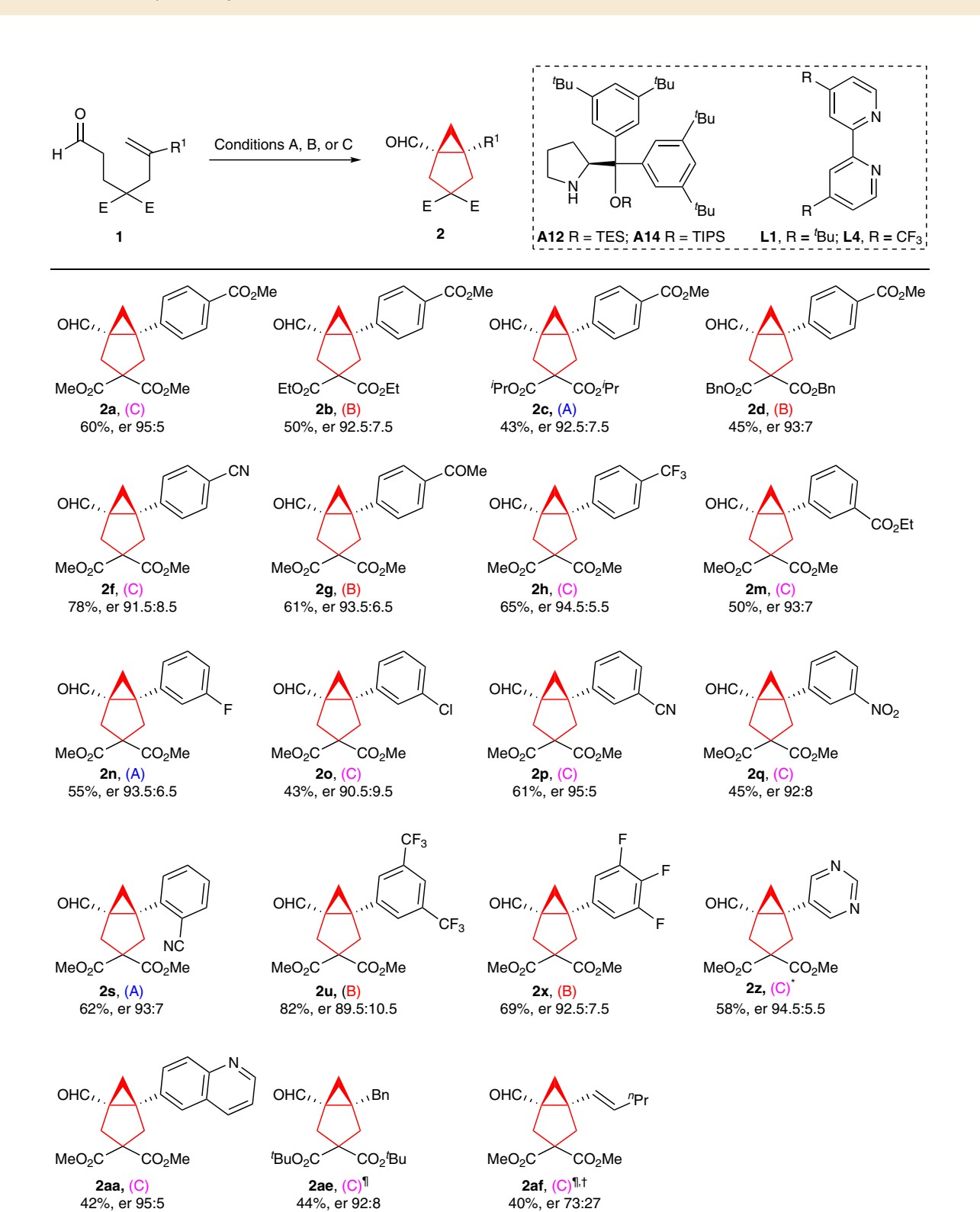

2a, (C)
60%, er 95:5

2b, (B)
50%, er 92.5:7.5

2c, (A)
43%, er 92.5:7.5

2d, (B)
45%, er 93:7

2f, (C)
78%, er 91.5:8.5

2g, (B)
61%, er 93.5:6.5

2h, (C)
65%, er 94.5:5.5

2m, (C)
50%, er 93:7

2n, (A)
55%, er 93.5:6.5

2o, (C)
43%, er 90.5:9.5

2p, (C)
61%, er 95:5

2q, (C)
45%, er 92:8

2s, (A)
62%, er 93:7

2u, (B)
82%, er 89.5:10.5

2x, (B)
69%, er 92.5:7.5

2z, (C)*
58%, er 94.5:5.5

2aa, (C)
42%, er 95:5

2ae, (C)¶
44%, er 92:8

2af, (C)¶,†
40%, er 73:27

Conditions A: CuI (20 mol%), **L1** (10 mol%), **A12** (20 mol%), F-BI-OH (2 equiv.), MeCN (0.1 M), 20 °C, 48 h; Conditions B: CuI (20 mol%), **L1** (10 mol%), **A12** (20 mol%), DF-BI-OH (2 equiv.), $n$-Bu$_4$NI (20 mol%), MeCN (0.1 M), 10 °C, 72 h; Conditions C: CuI (20 mol%), **L1** (10 mol%), **A14** (20 mol%), DF-BI-OH (2 equiv.), $n$-Bu$_4$NI (20 mol%), MeCN (0.1 M), 10 °C, 72 h; Er was determined by chiral stationary HPLC; the isolated yield was shown
* CuI (30 mol%), **L1** (15 mol%), **A14** (25 mol%), DF-BI-OH (2 equiv.), $n$-Bu$_4$NI (25 mol%), MeCN (0.1 M), 20 °C, 48 h
¶ CuI (30 mol%), **L4** (15 mol%), **A14** (25 mol%), DF-BI-OH (2 equiv.), $n$-Bu$_4$NI (25 mol%), MeCN (0.1 M), 10 °C, 72 h
† Reaction was run at 20 °C for 62 h

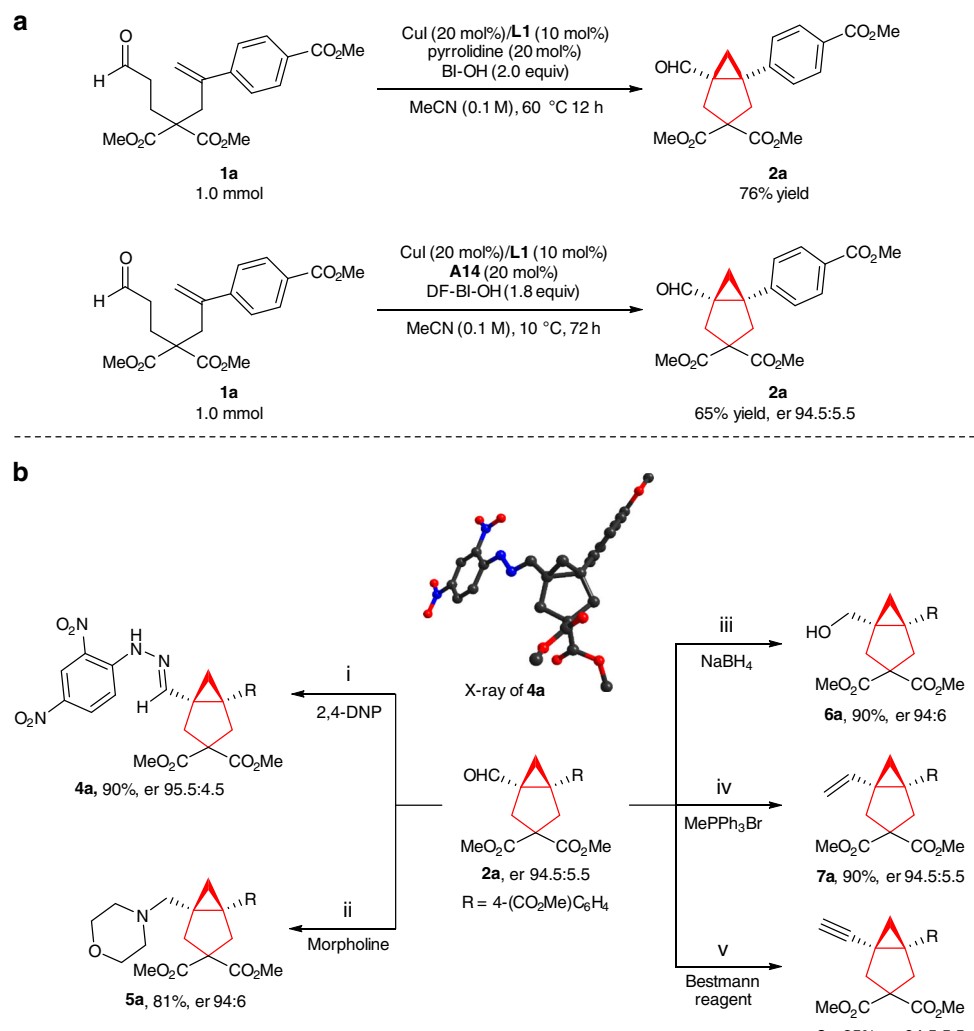

**Fig. 2** Synthetic application. **a** Large-scale preparation of **2a**. **b** Diverse transformations (**i**) 2,4-DNP, TsOH, DCM, rt, (**ii**) morpholine, NaBH(OAc)$_3$, DCE, 50 °C, (**iii**) NaBH$_4$, MeOH, rt, (**iv**) methyltriphenylphosphonium bromide, $^t$BuOK, THF, reflux; yield was based on recovered starting material, (**v**) Bestmann reagent = [dimethyl(acetyldiazomethyl)phosphonate], K$_2$CO$_3$, MeOH, rt. 2,4-DNP (2,4-dinitrophenyl)hydrazine, DCE dichloroethane

transformation have excellent functional group tolerance, particularly for the ones that are usually incompatible in radical-involved reactions under harsh conditions (halides, carbonyl groups, or NO$_2$). In addition, 3,5-disubstituted and 3,4,5-trisubstituted substrates were also suitable for this reaction, delivering the bicyclo[3.1.0]hexane products **2u** and **2x** in 89.5:10.5 and 92.5:7.5 er, respectively. To further investigate the reaction scope, we tested the use of heteroarene substituted alkene as the substrate. To our delight, the reaction gave the desired products **2z** and **2aa** in high enantioselectivity. Noteworthy is that alkyl- and alkenyl-substituted alkenes could also be employed in the reaction to give the desired products **2ae** and **2af** in moderate yields with moderate to good enantioselectivity, which is currently under further optimization in our laboratory. These features indicate that this general asymmetric cyclopropanation reaction exhibits broad substrate scope covering distinctly aromatic, heteroaromatic, alkenyl, alkyl-substituted alkenes, which are much less effective in previous radical-initiated asymmetric difunctionalization of alkenes[44–46].

reaction efficiency and enantioselectivity, indicating this protocol should be potential for large-scale chemical production of chiral bicyclo[3.1.0]hexanes. The other important aspect of this current methodology is that structurally diverse bicyclo[3.1.0]hexane skeletons containing vicinal all-carbon quaternary stereocenters are efficiently constructed. Consequently, the resultant functionalized compounds can serve as pivotal intermediates for easy access to other valuable chiral compounds (Fig. 2b). For example, the conversion of **2a** to its hydrazone derivative **4a** was achieved in 90% yield and the absolute configuration of **2a** was also unambiguously determined to be 1 S,5 S by X-ray crystal-structure analysis of **4a**. Reductive amination with morpholine gave amine **5a** in 81% yield. The corresponding alcohol **6a** could be obtained in 90% yield through the reduction with NaBH$_4$. Wittig reaction of **2a** installed an olefin group (**7a**) at the bridgehead. Alkyne functionality (**8a**) was also readily accessible using Bestmann's reagent. Notably, there was no loss of enantioselectivity during the above transformations and some derivatives are not readily accessible through traditional processes.

**Diverse synthetic application.** To illustrate the synthetic applicability of this transformation, a large-scale preparation of **2a** was performed. As shown in Fig. 2a, there was no change in

**Mechanism investigation.** To gain some insights into the reaction mechanism, a series of control experiments were conducted. First, the radical trapping experiment using TEMPO as a radical

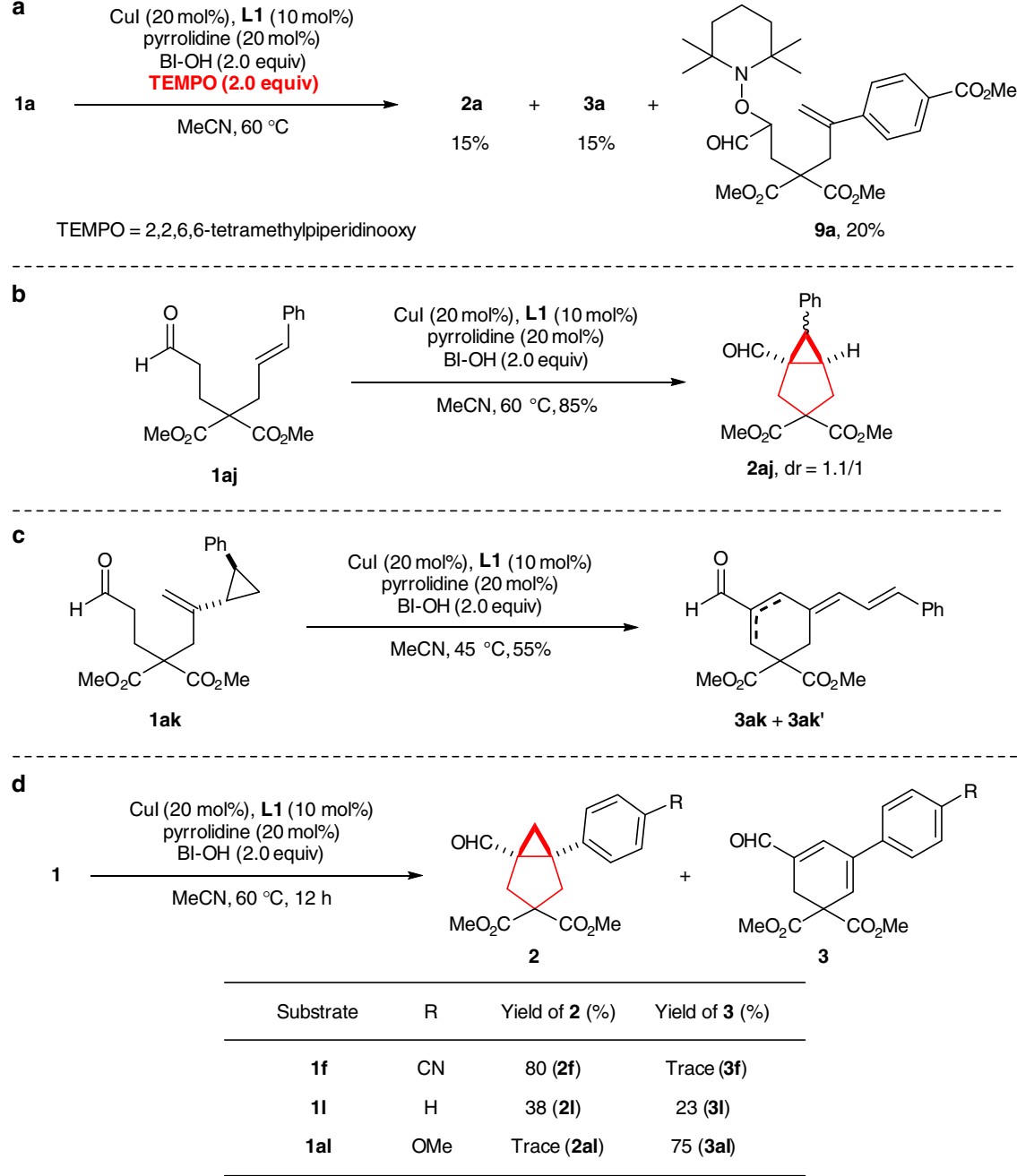

**Fig. 3** Control experiments on the radical pathway. **a** The control experiment using a radical scavenger. **b** Stepwise radical cyclopropanation. **c** The radical clock experiment. **d** The effect of para-substituent on the formation of cyclohexadiene

scavenger under the standard reaction conditions demonstrated significant reaction inhibition (Fig. 3a). The TEMPO-trapped adduct **9a** was detected in the transformation. This result supported a radical mechanism starting from the formation of α-alkyl radical of iminium ion[63]. Second, *E*-alkene substrate **1aj** led to **2aj** as a 1.1:1 mixture of diastereomers under the standard conditions (Fig. 3b). The loss of alkene stereochemistry during the reaction ruled out a potential copper-mediated concerted cyclopropanation mechanism. Third, the radical clock experiment on substrate **1ak** bearing a cyclopropanyl-substituted alkene moiety yielded products **3ak** and **3ak′** as an inseparable mixture in 55% yield (Fig. 3c), supporting a radical *6-endo-trig* cyclization leading to tertiary radical **E** (see the overall mechanism below). Fourth, the formation of side 1,3-cyclohexadiene product **3** was

favored on substrates containing an electron-rich substituent at *para* position of the alkenyl aryl ring (Fig. 3d), revealing that the cyclopropanation unlikely involves a carbocation intermediate, such as **G** in (see the overall mechanism below). All these results supported a stepwise radical cyclization mechanism for the current cyclopropanation reaction.

Preferences for *5-exo-trig* and *6-endo-trig* radical cyclization pathways have been reported for alkenes with different substitution patterns. To investigate the cyclization pathway of our current reaction, we have prepared aryl ketone **1am** with an aryl-substituted geminal alkene moiety and aryl ketone **1an** possessing a terminal alkene group (Fig. 4). Under conditions similar to the standard conditions, **1am** afforded the **3am** in 40% yield, possibly deriving from sequential *6-endo-trig* radical cyclization to form

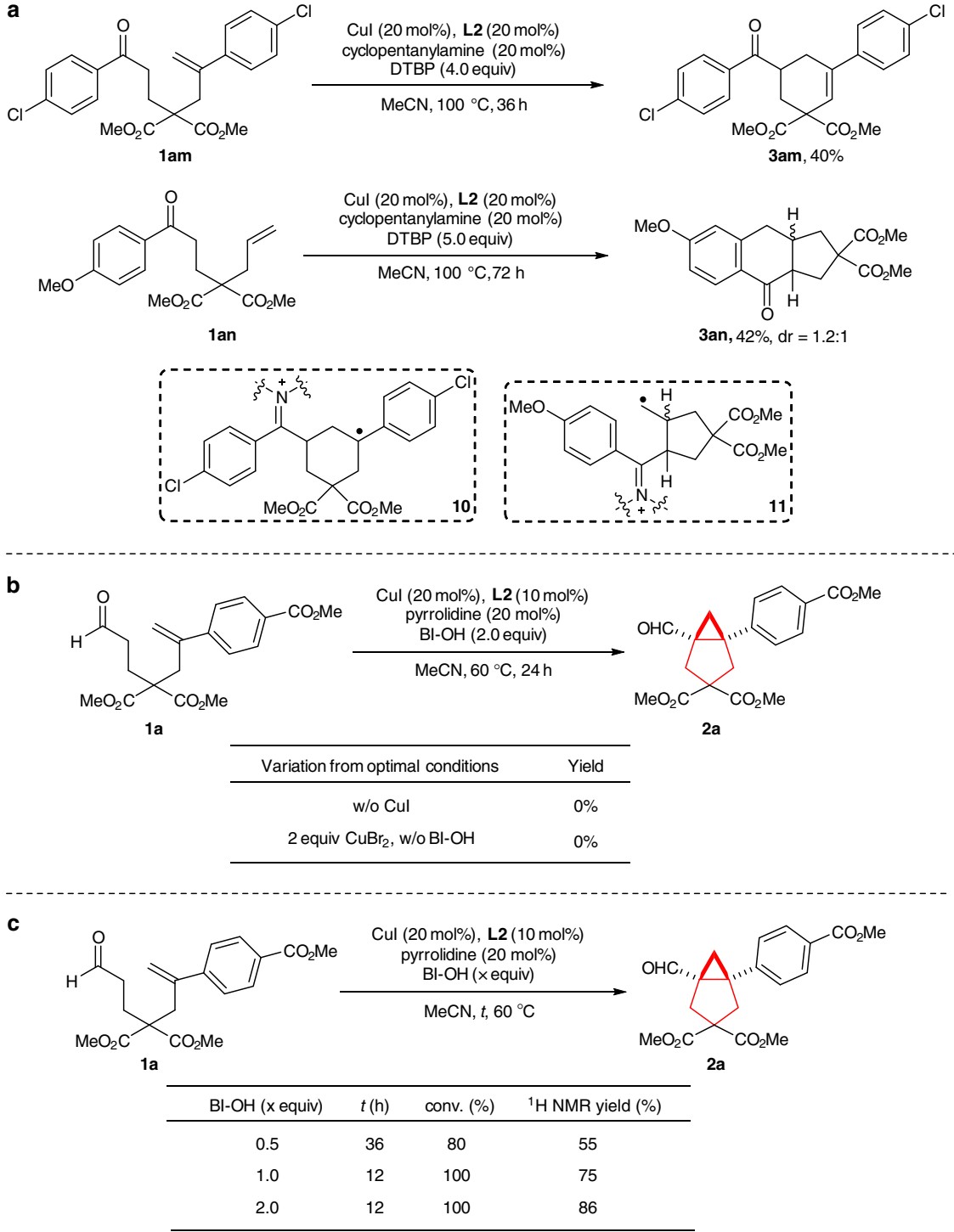

**Fig. 4** Mechanism study. **a** Preferences for *5-exo-trig* and *6-endo-trig* radical cyclization pathways. **b**, **c** Control experiments on catalyst and oxidant

intermediate **10** and further oxidation (Fig. 4a). Furthermore, the above-mentioned radical clock experiment and the formation of 1,3-cyclohexadiene product **3** also support the *6-endo-trig* radical cyclization pathway (Fig. 3c, d). However, **1an** led to tricyclic compound **3an**, possibly via sequential *5-exo-trig* cyclization and attack of the aryl ring by the resultant primary radical **11** (Fig. 4a). These facts are in accordance with literature precedents reporting different cyclization preferences of aryl-substituted geminal alkene and mono-substituted terminal alkene[64–67]. Thus, the exact radical cyclization pathway of our reaction should

depend on the substitution pattern of the alkene (see the proposed overall mechanism below for a brief summary).

Besides, the control reactions conducted in the absence of either the copper catalyst or BI-OH (two equivalents of CuBr$_2$ was used instead) did not provide either **2a** or **3a** (Fig. 4b). This result confirmed that the combination of BI-OH and copper catalyst was essential for this reaction. Further study on the stoichiometry of BI-OH disclosed that one equivalent of BI-OH was sufficient for a full conversion of **1a** (Fig. 4c) and thus the oxidant participated two SET processes during the reaction[68].

**Fig. 5** A plausible reaction mechanism. Mono-substituted terminal alkene substrate and 1,2-disubstituted internal alkene substrate bearing a phenyl group prefer the *5-exo-trig* cyclization pathway while 1,1-disubstituted alkene substrate bearing an aryl or an alkenyl group favors the *6-endo-trig* cyclization pathway

Based on the above observations and previous studies[32,33,55–59], a tentative mechanism for this transformation is proposed (Fig. 5). Initially, alkenyl aldehyde **1** was converted, via condensation of aldehyde with amine catalyst, to the enamine intermediate **A**, which could undergo a SET process with BI-OH or Cu(II) generated in situ to form α-alkyl radical of iminium ion **B**[55–59,63]. Depending on the substitution pattern of the alkene moiety, two cyclization pathways may predominate, respectively. For terminal alkene substrate **2ab** and **2ah** and internal alkene substrate **2aj**, the *5-exo-trig* cyclization pathway is kinetically favored[64,65]. However, for aryl and alkenyl-substituted alkene substrates, the *6-endo-trig* pathway is likely favored due to the stabilization of the resultant radical via conjugation[66,67]. As for alkyl-substituted alkene substrates, the ratio between these two pathways may vary depending mainly on the steric bulkiness of these alkyl substituents[66,67]. The radical intermediates **C** and **E** are likely stabilized via formation of organocopper species **C′** and **E′**, respectively[69]. The subsequent cyclopropanation occurs most likely through radical *3-exo-trig* cyclization followed by further oxidation of resultant aminoalkyl radical to iminium intermediates **D** and **F**. Direct intramolecular $S_N2$ displacement of organocopper species by the enamine moiety cannot be ruled out for the formation of **D** at present[70]. Finally, hydrolysis of the resultant iminium gives rise to product **2**. Carbocation **G** may be formed from **E** or **E′** by oxidation or heterolytic cleavage of the C–Cu bond, respectively, which upon deprotonation and further oxidation leads to side product diene **3**.

## Discussion

We have developed a general intramolecular radical cyclopropanation of unactivated alkenes with simple α-methylene group of aldehydes as a C1 source, which provides facile access to bicyclo [3.1.0]hexane skeletons with excellent efficiency, broad substrate scope, and excellent functional group tolerance. Furthermore, a catalytic asymmetric radical α-cyclopropanation of aldehydes in the presence of a Cu(I)/chiral secondary amine cooperative system has been achieved. This process provides an attractive and promising approach to fundamental yet synthetically formidable chiral bicyclo[3.1.0]hexane skeletons bearing two highly congested vicinal all-carbon quaternary stereocenters in good yields with high levels of enantioselectivity, featuring mild reaction conditions, a remarkably broad substrate scope covering diverse aromatic, heteroaromatic, alkenyl, and alkyl-substituted geminal alkenes with excellent functional group tolerance. Noteworthy is that simple α-methylene group of an aldehyde served as a good C1 source in this efficient asymmetric [2 + 1] cycloaddition, which constitutes an ideal strategy for cyclopropanation with

respect to ready accessibility of starting materials, operation safety, and atom economy. The use of cyclic hypervalent iodine (III) reagent as the single electron oxidant plays an important role in the context of this transformation. The unique bridgehead aldehyde functionality was converted to various useful chiral scaffolds. The realization of this transformation might provide useful insight for addressing the challenges in related radical-initiated asymmetric difunctionalization of unactivated alkenes with dual-catalytic system.

## Methods

**Racemic radical intramolecular cyclopropanation**. To a flame-dried Schlenk tube equipped with a magnetic stir bar were added 1 (0.2 mmol), CuI (7.6 mg, 20 mol%), ligand (10 mol%), and BI-OH (108 mg, 0.4 mmol). The tube was evacuated and backfilled with argon for three times. Pyrrolidine (3.3 μL, 20 mol%) and freshly degassed acetonitrile (2.0 mL) was added via syringe. The tube was stirred at 60 °C for 12–24 h until TLC (Thin-layer Chromatography) monitored the full completion of starting material. After completion, solvent was removed under reduced pressure, and the residue was diluted with ethyl acetate (15 mL), washed with saturated NaHCO₃ solution, then washed with brine, dried with MgSO₄, filtered, and concentrated. Flash chromatography (petroleum ether/ethyl acetate = 10/1–5/1) gave the corresponding products 2.

**Asymmetric radical intramolecular cyclopropanation**. To a flame-dried Schlenk tube equipped with a magnetic stir bar were added 1 (0.1 mmol), CuI (3.8 mg 20 mol%), L1 (2.6 mg, 10 mol%), DF-BI-OH (60 mg, 0.2 mmol), *n*-Bu₄NI (7.4 mg, 20 mol%), and Amine 14 (12.8 mg, 20 mol%). The tube was evacuated and backfilled with argon for three times, the freshly degassed dry acetonitrile (1.0 mL) was added via syringe. The tube was stirred at 10 °C for 72 h. After completion, solvent was removed under reduced pressure, and the residue was diluted with saturated NaHCO₃ solution, then extracted with ethyl acetate. The organic layer was washed with brine, dried with MgSO₄, filtered, and concentrated. Flash chromatography (petroleum ether/ethyl acetate = 10/1–5/1) gave the corresponding products 2.

For nuclear magnetic resonance and high-performance liquid chromatography spectra, see Supplementary Figures.

**Data availability**. The X-ray crystallographic coordinates for structure reported in this article have been deposited at the Cambridge Crystallographic Data Centre (CCDC), under deposition number CCDC 1515426 ((S,S)-4a). The data can be obtained free of charge from The Cambridge Crystallographic Data Centre via http://www.ccdc.cam.ac.uk/data_request/cif. Any further relevant data are available from the authors upon reasonable request.

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

### Acknowledgements

Financial support from the National Natural Science Foundation of China (nos. 21722203, 21702093, and 21572096), and Shenzhen special funds for the development of biomedicine, internet, new energy, and new material industries (JCYJ20170412152435366 and JCYJ20170307105638498) is greatly appreciated.

### Author contributions

L.Y, and Q.-S.G. performed experiments. Y.T, X.M, and G.-C.C. helped with characterizing all new compounds. X.-Y.L. conceived and directed the project and wrote the paper.

### Additional information

**Competing interests:** The authors declare no competing financial interests.

