## [Peer Review File · Nature Communications]

Reviewer #1 (Remarks to the Author):

The manuscript by Liu and coworkers describes a secondary amine/Cu catalyzed intramolecular cyclopropanation of olefinic aldehydes. This process is believed to proceed via an imino alpha-radical that reacts with a tethering alkene. Substituted bicyclo[3.1.0]hexane analogues were prepared in one step with moderate to good yield. The authors subsequently accomplished the asymmetric version of this transformation by using a modified Hayashi-Jørgensen's diarylprolinol silyl ether catalyst. Both yield and ee were synthetically useful. Direct asymmetric cyclopropanation between a CH₂ and an olefin is very challenging. As the authors mentioned, classical methods for cyclopropanes primarily relied on metalcarbene intermediates that require pre-functionalization of starting materials. The submitted work is an excellent example that the native alpha-CH₂ of an aldehyde can undergo cyclopropanation without any pre-activation. Publication is recommended pending the following revisions.

1. The overall transformation is analogous to the one described by Huang in Ref. 27. A short discussion is needed in the introduction section to compare the difference.
2. Figure 5 is confusing. R1 or R2 is missing from several intermediates. In addition, the overall mechanistic scheme need to be revised for better clarity. It seems E leads to F (6-endo-trig??) and E' leads to G, which is not correct.
3. What is the role of Cu? The authors mentioned stabilization of radical C by forming an organocopper C' species (Fig. 5). Isn't C more reactive towards cyclopropanation? Why this radical needs to be stabilized? Could copper be the actually oxidant and got turned over by DF-BI-OH?
4. Did the authors try using stoichiometric copper and lose DF-BI-OH? MacMillan reported copper salts are effective oxidants for converting enamines to imino radicals.
5. It is unclear how 5-exo-trig and 6-endo-trig cyclization is controlled. A short summary is desired to describe which type of substrates favors 5-exo-trig and which favors 6-endo-trig.

Reviewer #2 (Remarks to the Author):

This communication by the Liu group describes a highly original and powerful synthesis of formyl bicyclo[3.1.0]hexanes, bearing two quaternary centers at the ring junction. The strategy looks quite general and can be adapted to an asymmetric version which gives high value to the proposed catalytic process. The mechanistic study looks sound.

I recommend publication in Nature Communications after addressing the following points:

- The seminal work by Sibi on the organocatalyzed alpha-oxygenation of aldehydes (J. Am. Chem. Soc. 2007, 129, 4124) should be cited around references 55-59.
- On page 6, it is claimed that "it was interesting to find that substrates bearing electron-donating substituents (Me, or OMe) or electron-neutral (H) arene rings at different positions were all suitable for the reaction". This is not true when you consider the reaction of methoxy precursor of page 13, Figure 3d (which numbering seems wrong by the way).
- There should be more examples of substrates with a tether different than a malonate group. What about O-tethered or N-tethered substrates? Do they work?
- The "Diverse synthetic application" section of page 10 is not really fascinating.
- On Figure 5, at which oxidation level is the copper which intercepts the radical species? Assuming it is copper(II), a possible mechanism for the formation of the diene 3 could be the formation of a transient copper(III) intermediate which undergoes beta-hydride elimination to generate the alkene. After reductive elimination a copper(I) species is created. This would be consistent with Barry Snider's mechanism proposal with the Mn(III)/Cu(II) mixture (see his Chem. Rev.).

- In the SI, the ^{19}F NMR of compounds containing fluorine should be given.
- Several typos, for instance:
- On page 10, correct "hydrozone".
 - On page 14, line 301, it should be "depends".

Our Responses to the Comments of the Reviewers

Reviewer 1

Comment 1: *The manuscript by Liu and coworkers describes a secondary amine/Cu catalyzed intramolecular cyclopropanation of olefinic aldehydes. This process is believed to proceed via an imino alpha-radical that reacts with a tethering alkene. Substituted bicyclo[3.1.0]hexane analogues were prepared in one step with moderate to good yield. The authors subsequently accomplished the asymmetric version of this transformation by using a modified Hayashi-Jørgensen's diarylprolinol silyl ether catalyst. Both yield and ee were synthetically useful. Direct asymmetric cyclopropanation between a CH₂ and an olefin is very challenging. As the authors mentioned, classical methods for cyclopropanes primarily relied on metalcarbene intermediates that require pre-functionalization of starting materials. The submitted work is an excellent example that the native alpha-CH₂ of an aldehyde can undergo cyclopropanation without any pre-activation. Publication is recommended pending the following revisions*

Our response: We very much appreciate these comments of the reviewer and sincerely thank him for recommending publication of the work.

Comment 2: *The overall transformation is analogous to the one described by Huang in Ref. 27. A short discussion is needed in the introduction section to compare the difference.*

Our response: We thank the reviewer for bringing this important issue to our attention and feel sorry for our negligence. We have added a short discussion for Huang's seminal work and have compared the differences between our work and Huang's one in the revised manuscript. "Noteworthy is that Huang and co-workers has recently reported an asymmetric intramolecular α -cyclopropanation of alkenyl aldehydes with the in-situ generated α -iodoaldehyde as a donor/acceptor carbene mimetic, invoking a stepwise double electrophilic alkylation cascade through 5-*exo-trig* cyclization.²⁷ In this reaction, the bis-alkyl substituents at the double bond were essential for implementing the enantioselective reaction, possibly due to indispensable formation of carbocation intermediates, with an exceptionally stoichiometric amount of chiral secondary amine as the promoter."

Comment 3: *Figure 5 is confusing. R1 or R2 is missing from several intermediates. In addition, the overall mechanistic scheme need to be revised for better clarity. It seems **E** leads to **F** (6-*endo-trig*??) and **E'** leads to **G**, which is not correct.*

Our response: We thank the reviewer for pointing out this issue and apologize for such an unclear mechanism description. We have revised the overall mechanistic scheme in the revised manuscript (as shown in Scheme 1). **E** and **E'** are two forms of intermediates, which could be converted to intermediate **F** via intramolecular cyclopropanation or intermediate **G** via further oxidation.

Scheme 1. Comparison between previous mechanism (left) and revised one (right)

Comment 4: What is the role of Cu? The authors mentioned stabilization of radical C by forming an organocopper C' species (Fig. 5). Isn't C more reactive towards cyclopropanation? Why this radical needs to be stabilized? Could copper be the actually oxidant and got turned over by DF-BI-OH?

Our response: We thank the reviewer for the valuable suggestion. The role of Cu in this transformation is believed to have two aspects. On one hand, copper might function as the actual oxidant, as shown in Fig. 5, which gets turned over by cyclic hypervalent iodine(III) reagent. On the other hand, copper might stabilize alkyl radical species through formation of intermediate C' and intermediate E' in Fig. 5 to facilitate the desired cyclization (also see ref. 69 in the text). Intermediate C, in principle, may be more reactive towards cyclopropanation. However, this highly reactive alkyl radical species is prone to many side reactions, thus resulting in poor chemoselectivity. The organocopper C' species may function as either the immediate reactant or a sequestered pool for radical C, in which way it may afford the desired product in a high chemoselectivity manner.

Comment 5: Did the authors try using stoichiometric copper and lose DF-BI-OH? MacMillan reported copper salts are effective oxidants for converting enamines to imino radicals.

Our response: We thank the reviewer for the valuable advice. Actually, we have performed the reaction using 2.0 equivalents of CuBr₂ in the absence of BI-OH, while no desired product was obtained (the result was listed in Figure 4b in the text). This result also indicates that the role of copper salts was not only as the possible oxidant in our transformation although they are effective oxidants for converting enamines to imino radicals in MacMillan's report.

b) The control experiments in the absence of copper catalyst or oxidant

Figure 4b Control experiments on catalyst and oxidant.

Comment 6: It is unclear how 5-exo-trig and 6-endo-trig cyclization is controlled. A short summary is desired to describe which type of substrates favors 5-exo-trig and which favors 6-endo-trig.

Our response: We thank the reviewer for pointing out this issue and feel sorry for not emphasizing the summary in the text (at the end of line 302, page 14). We added the note “(see the proposed overall mechanism below for a brief summary)” in the end. In overall mechanism description, we have drawn a brief summary on substrate-controlled 5-exo-trig and 6-endo-trig cyclization: “Depending on the substitution pattern of the alkene moiety, two cyclization pathways may predominate, respectively. For terminal alkene substrate **2ab** and **2ah** and internal alkene substrate **2aj**, the 5-exo-trig cyclization pathway is kinetically favored.^{64,65} However, for aryl and alkenyl-substituted alkene substrates, the 6-endo-trig pathway is likely favored due to the stabilization of the resultant radical via conjugation.^{66,67} As for alkyl-substituted alkene substrates, the ratio between these two pathways may vary depending mainly on the steric bulkiness of these alkyl substituents.^{66,67}”

Reviewer 2

Comment 1: This communication by the Liu group describes a highly original and powerful synthesis of formyl bicyclo[3.1.0]hexanes, bearing two quaternary centers at the ring junction. The strategy looks quite general and can be adapted to an asymmetric version which gives high value to the proposed catalytic process. The mechanistic study looks sound. I recommend publication in *Nature Communications* after addressing the following points:

Our response: We very much appreciate these comments of the reviewer and sincerely thank you for recommending publication of the work after minor revision.

Comment 2: The seminal work by Sibi on the organocatalyzed alpha-oxygenation of aldehydes (*J. Am. Chem. Soc.* 2007, 129, 4124) should be cited around references 55-59.

Our response: We thank the reviewer for pointing out this issue and feel sorry for missing such an important reference. We have added the reference as ref 59 in the revised text.

Comment 3: On page 6, it is claimed that “it was interesting to find that substrates bearing electron-donating substituents (Me, or OMe) or electron-neutral (H) arene rings at different positions were all suitable for the reaction”. This is not true when you consider the reaction of

methoxy precursor of page 13, Figure 3d (which numbering seems wrong by the way).

Our response: We thank the reviewer for the valuable advice. We have revised the description in the updated manuscript to “It was interesting to find that substrates bearing electron-donating substituents (*ortho*-Me, or *meta*-OMe) or electron-neutral (H) arene rings were all suitable for the reaction to produce the desired products **2i**, **2r** and **2v** in moderate to good yields, while with a small amount of unexpected 1,3-cyclohexadiene derivatives.”

In Figure 3d, entry 3, the number of *para*-OMe substituted substrate **1al** is right. We are sorry for confusing you.

Figure 3d The effect of *para*-substituent on the formation of cyclohexadiene

Comment 4: There should be more examples of substrates with a tether different than a malonate group. What about *O*-tethered or *N*-tethered substrates? Do they work?

Our response: We thank the reviewer for pointing out this issue and thus, have prepared *O*-tethered and *N*-tethered substrates accordingly. However, both of them gave disappointing results. For *N*-tethered substrate **1ao**, under conditions listed as entries 2 and 9 in Table 1, the desired product **2ao** could be obtained together with the formation of **4ao** as an inseparable mixture. When other solvents were employed (entries 3 and 4) or pyrrolidine (entries 5–8) was used as catalyst, no desired product was detected. After reduction with NaBH₄, the thus obtained alcohol product **5ao** could be separated from **4ao** (Scheme 2, top). When we subjected **1ao** to the optimal asymmetric conditions, better yield was observed compared to those obtained under racemic conditions, but only 10% enantioselectivity was obtained (Scheme 2, bottom).

For *O*-tethered substrate **1ap**, under conditions listed as entries 1-6 in Table 2, the desired product **2ap** could not be detected while with the formation of **4ap** as a major byproduct, probably resulting from disproportion of aldehyde substrate.

Thus, we are unfortunately unable to provide publishable results for substrates of these two types for now and are still making efforts to improve the reaction efficiency and enantioselectivity in our laboratory. The corresponding results will be disclosed in due report.

Table 1. Screening of Reaction Conditions for *N*-Tethered Substrate

$\text{1ao} \xrightarrow[\text{MeCN, 40 } ^\circ\text{C, 24 h}]{\text{CuI (20 mol\%)/Ligand (x mol\%)\ amine (y mol\%)/oxidant (z equiv)}} \text{2ao} + \text{4ao}$

L1 **L2** **Amine I** **Amine II**

entry	Ligand (y mol%)	amine (z mol%)	oxidant (z equiv)	yield of 2ao (2ao/4ao)
1*	L2 (20)	Amine I (20)	DTBP (5.0)	0 (0/50)
2	L2 (20)	Amine I (20)	BI-OH (2.0)	15 (15/30)
3 [#]	L2 (20)	Amine I (20)	BI-OH (2.0)	messy complex
4 ^{&}	L2 (20)	Amine I (20)	BI-OH (2.0)	messy complex
5	L2 (10)	pyrrolidine (30)	BI-OH (2.0)	messy complex
6	L2 (10)	pyrrolidine (40)	BI-OH (2.0)	messy complex
7 ^ψ	L1 (10)	pyrrolidine (40)	BI-OH (2.0)	messy complex
8	L1 (10)	pyrrolidine (40)	BI-OH (2.0)	messy complex
9 [§]	L1 (20)	Amine II (40)	BI-OH (2.0)	12 (12/24)

* CuI (10 mol%) was used and reaction temperature is 80 °C; # Methyl *tert*-butyl ether was used as solvent; & CHCl₃ was used as solvent; ^ψ Tetrabutylammonium iodide (20 mol%) was added and reaction temperature is room temperature; § CuI (40 mol%) and tetrabutylammonium iodide (30 mol%) were used and reaction temperature is room temperature.

**Scheme 2** Reduction of **2am** for separation (top) and asymmetric trial for **1am** (bottom)

Table 2. Screening of Reaction Conditions for *O*-Tethered Substrate
entry	Ligand (y mol%)	amine (z mol%)	oxidant (z equiv)	yield of 2ap (2ap/4ap)
1*	L2 (10)	pyrrolidine (20)	BI-OH (2.0)	0 (0/35)
2 [#]	L2 (10)	pyrrolidine (20)	BI-OH (2.0)	0 (0/32)
3 ^{&}	L2 (10)	pyrrolidine (20)	BI-OH (2.0)	0 (0/28)
4 ^{&} §	L1 (10)	pyrrolidine (30)	DFBI-OH (2.0)	0 (0/36)
5 ^{&} § ε	L1 (10)	pyrrolidine (30)	DFBI-OH (2.0)	0 (0/30)
6 ^{&} § ε	L1 (10)	A12 (20)	DFBI-OH (2.0)	0 (0/20)

*The reaction temperature is 60 °C; # The reaction temperature is 80 °C; & Tetrabutylammonium iodide (20 mol%) was added; § The reaction temperature is 10 °C, 36 h; ε 4-fluorobenzoic acid (20 mol%) was added.

Comment 5: The "Diverse synthetic application" section of page 10 is not really fascinating.

Our response: We sincerely thank the reviewer for suggesting us conduct further study on this aspect. Thus, we are currently making efforts towards more diverse synthetic applications of this useful methodology in our lab. Unfortunately, we are unable to provide the results in this manuscript for now, which may be disclosed in due report as soon as possible.

Comment 6: On Figure 5, at which oxidation level is the copper which intercepts the radical species? Assuming it is copper(II), a possible mechanism for the formation of the diene 3 could be the formation of a transient copper(III) intermediate which undergoes beta-hydride elimination to generate the alkene. After reductive elimination a copper(I) species is created. This would be consistent with Barry Snider's mechanism proposal with the Mn(III)/Cu(II) mixture (see his *Chem. Rev.*).

Our response: We sincerely thank the reviewer for these valuable suggestions. According to literature, the alkyl radical generated is much likely intercepted by Cu(II) (*J. Am. Chem. Soc.* **1965**, 87, 4855; *J. Am. Chem. Soc.* **1968**, 90, 4616; *J. Am. Chem. Soc.* **2014**, 136, 6011; *Sci. Rep.* **2017**, 7, 43579). However, we cannot absolutely exclude the potential interception by Cu(I) (*Isr. J. Chem.* **1990**, 30, 361). Accordingly, we have explicitly added the oxidation state in the mechanistic scheme (Fig. 5 in the revised manuscript).

Regard to the mechanism for the formation of alkene side products, on the basis of Kochi's work (*J. Am. Chem. Soc.* **1968**, 90, 4616), the corresponding organocopper(III) species generated from primary and secondary alkyl radicals intercepted by Cu(II) should readily undergo concerted oxidative elimination without a carbocation intermediate, i.e., the one suggested by the reviewer as well as involved in Prof. Snider's mechanism proposal (*Chem. Rev.* **1996**, 96, 339). However, for tertiary alkyl radicals, particularly for tertiary arylalkyl radicals such as those involved in our reaction, their oxidation by Cu(II) via formation of organocopper(III) intermediate leading to relatively stable carbocation should be facile and subsequent E1-like deprotonation should provide

the final alkene side product (*J. Org. Chem.* **1968**, 33, 83). Such a mechanism has also been invoked by other related processes (e.g., *J. Am. Chem. Soc.* **2014**, 136, 6011). Therefore, although we are unable to absolutely exclude the mechanism kindly suggested by this reviewer, we prefer the one drawn in our manuscript due to the reason mentioned above.

Comment 7: In the SI, the ^{19}F NMR of compounds containing fluorine should be given.

Several typos, for instance:

- On page 10, correct "hydrozone".

- On page 14, line 301, it should be "depends".

Our response: We sincerely thank the reviewer for bringing these issues to our attention and apologize for our negligence. We have already provided the corresponding ^{19}F NMR spectra of compounds containing fluorine and corrected these typos according to his valuable suggestions.

5ao, ^1H NMR (400 MHz, Chloroform-*d*) δ 7.97 (d, J = 8.1 Hz, 2H), 7.74 (d, J = 8.1 Hz, 2H), 7.38 (d, J = 8.1 Hz, 2H), 7.31 (d, J = 8.1 Hz, 2H), 3.92 (s, 3H), 3.80 (dd, J = 12.4, 9.6 Hz, 2H), 3.49 – 3.43 (m, 2H), 3.36 – 3.33 (m, 2H), 2.49 (s, 3H), 1.26 (d, J = 5.5 Hz, 1H), 1.17 (d, J = 5.4 Hz, 1H). ^{13}C NMR (100 MHz, CDCl_3) δ 166.61, 143.73, 142.03, 133.74, 129.93, 129.79, 129.51, 129.36, 127.55, 62.84, 56.09, 52.17, 51.85, 35.95, 35.35, 21.59, 16.34. **HRMS** (ESI) m/z calcd. for $\text{C}_{21}\text{H}_{24}\text{O}_5\text{NS}$ $[\text{M}+\text{H}]^+$ 402.1369, found 402.1375.

4ap, ^1H NMR (400 MHz, Chloroform-*d*) δ 7.61 – 7.56 (m, 8H), 5.62 (d, J = 4.4 Hz, 2H), 5.44 (d, J = 5.6 Hz, 2H), 4.40 (s, 2H), 4.36 (s, 2H), 4.14 (t, J = 6.4 Hz, 2H), 3.76 (t, J = 6.4 Hz, 2H), 3.53 (t, J = 6.4 Hz, 2H), 2.57 (t, J = 6.4 Hz, 2H), 1.92 – 1.86 (m, 2H). ^{13}C NMR (100 MHz, CDCl_3) δ 171.43, 143.25, 143.00, 142.23, 142.15, 126.41, 126.37, 125.27 (m, $J_{\text{C-F}}$ = 3.7 Hz), 124.27 (q, $J_{\text{C-F}}$ = 270.2 Hz), 116.68, 116.39, 72.91, 72.73, 66.53, 65.57, 61.74, 35.02, 28.92. **HRMS** (ESI) m/z calcd. for $\text{C}_{26}\text{H}_{27}\text{F}_6\text{O}_4$ $[\text{M}+\text{H}]^+$ 517.1808, found 517.1805.

y117-806POL/1

YL-17-806POL/2

YL-17-814PB/1

YL-17-814PB/2

Reviewer #1 (Remarks to the Author):

Most revision suggestions have been addressed by the authors. Publication is recommended.

Reviewer #2 (Remarks to the Author):

On the revised mechanism, Cu(III) intermediates (C' and E') have been drawn which implies that Cu(II) is drawn on the corresponding arrows.

Too bad the synthetic scope could not be extended.

Nevertheless, I recommend publication.